# Most Frequently Consumed Red/Processed Meat Dishes and Plant-Based Foods and Their Contribution to the Intake of Energy, Protein, and Nutrients-to-Limit among Canadians

**DOI:** 10.3390/nu14061257

**Published:** 2022-03-16

**Authors:** Mojtaba Shafiee, Naorin Islam, D. Dan Ramdath, Hassan Vatanparast

**Affiliations:** 1College of Pharmacy and Nutrition, University of Saskatchewan, Saskatoon, SK S7N 4Z2, Canada; mojtaba.shafiee@usask.ca (M.S.); naorin.islam@usask.ca (N.I.); 2Guelph Research and Development Centre, Agriculture and Agri-Food Canada, Guelph, ON N1G 5C9, Canada; 3School of Public Health, University of Saskatchewan, Saskatoon, SK S7N 4Z2, Canada

**Keywords:** red/processed meat dishes, plant-based foods, long-grain white rice, nutrients to limit, Canadian population

## Abstract

Using cross-sectional data from the 2015 Canadian Community Health Survey–Nutrition, we aimed to identify and characterize the top 10 most frequently consumed plant-based foods and red/processed meat dishes in the Canadian population. Plant-based foods and red/processed meat dishes categories included 659 and 265 unique food codes, respectively, from the Canadian Nutrient File. A total of 20,176 Canadian individuals aged ≥1 year were included in our analysis. The most frequently consumed plant-based food was “Cooked regular long-grain white rice”, which made a significant contribution to energy (12.1 ± 0.3%) and protein (6.1 ± 0.2%) intake among consumers. The most frequently consumed red/processed meat dish in Canada was “Cooked regular, lean or extra lean ground beef or patty”. Among red/processed meat dishes, “ham and cheese sandwich with lettuce and spread” made the most significant contribution to the intake of energy (21.8 ± 0.7%), saturated fat (31.0 ± 1.0%), sodium (41.8 ± 1.3%), and sugars (8.2 ± 0.5%) among the consumers. Ground beef is the most frequently consumed red/processed meat dish and white rice is the most frequently consumed plant-based food among Canadians. Red/processed meat dishes are major drivers of the excessive intake of nutrients-to-limit.

## 1. Introduction

In 2017, dietary risk factors were responsible for 255 million disability-adjusted life-years (DALYs) and 11 million deaths across 195 countries [1]. Globally, the three major dietary risk factors for mortality and DALYs were high intake of sodium, low intake of whole grains, and low intake of fruits [1]. Health Canada recommends the regular intake of nutritious foods (e.g., whole grains, fruit, vegetables) that are commonly found in dietary patterns such as Dietary Approaches to Stop Hypertension (DASH) and Mediterranean-style diets, which are known to be associated with beneficial effects on human health [2,3,4]. According to Canada’s new food guide (2019), consumption of nutritious foods often leads to low intakes of saturated fat (<10% of total energy intake), free sugars (<10% of total energy intake), or sodium (<2300 mg/day) [5]. While many animal-based foods are nutritious, Canada’s new food guide (2019) emphasizes the consumption of more plant-based foods because of the positive health effects associated with higher intakes of vegetables and fruit, nuts, soy protein, and dietary fiber [5]. Moreover, dietary shifts toward fewer animal-based foods and more plant-based foods could encourage lower intakes of processed meat (such as sausages, ham, and hot dogs), and foods high in saturated fat [5]. In this regard, Kirkpatrick et al. reported that red meat (beef, pork, lamb, and goat) mixed dishes are the largest contributors to saturated fat and sodium intake in the diet of Canadians [6]. Further, it has been found that rice (84%) is the most commonly consumed plant-based food, and beef (48.7%) is the most commonly consumed animal-based food among the Brazilian population aged 10 years or over [7]. A growing body of evidence suggests that consumption of red and processed meat is not only associated with poorer health outcomes but also with negative environmental impacts [8,9,10]. According to the EAT-Lancet Commission report, a diet with fewer animal source foods and rich in plant-based foods confers environmental benefits [11]. In this regard, a study conducted in the European Union (EU) showed that replacing 25–50% of animal-derived foods (i.e., eggs, dairy products and meat) in the EU with plant-based foods on a dietary energy basis would lead to less per capita use of cropland for food production (23%), a reduction in greenhouse gas emissions (25–40%), and a reduction in nitrogen emissions (40%) [12]. Examining the most frequently consumed plant-based foods and red/processed meat dishes and their contribution to dietary components gives us an overview of the current dietary habits of Canadians and provides insights into targets for interventions to support healthy eating patterns.

The primary objective of the present study was to identify the top 10 most frequently consumed plant-based foods and red/processed meat dishes in the Canadian population aged ≥1 year in 2015. We also aimed to (1) rank the top 10 plant-based foods and red/processed meat dishes based on their contribution to nutrients-to-limit, energy, and protein, and (2) determine the mean and percentage contribution of the top five plant-based foods and red/processed meat dishes to nutrients-to-limit, energy, and protein intake of consumers.

## 2. Materials and Methods

### 2.1. Study Population and Dietary Data Collection

Cross-sectional data from the Canadian Community Health Survey (CCHS)–Nutrition 2015 was used in this study. These survey data were collected from 20,487 individuals across ten provinces of Canada, excluding individuals living in Indigenous settlements and reserves, Canadian forces employees, and the institutionalized population [13,14]. The survey respondents provided 24 h dietary recall information as well as sociodemographic and supplement use information. For this study, we used day 1 of the 24 h recall. The dietary recall data included all foods and beverages consumed by participants within a 24 h period as well as frequency, time, location, and amount of food consumed. These also included information on food groups, nutrients, and eating patterns. The Automated Multiple Pass Method (AMPM) was used to derive dietary information [14]. This method is based on the United States Department of Agriculture (USDA) and it is an automated questionnaire that maximizes the survey respondent’s response to report and recall dietary intake in the last 24 h. A proxy interview was used to collect information from children aged 1–6 years under the supervision of parents or guardians. Children aged 6–11 years participated with their parental guidance, and respondents aged ≥12 years provided information using a non-proxy method. Detailed information on the survey design and methodology of the CCHS 2015–Nutrition can be found on the Statistics Canada website [13,14]. We accessed data at the Research Data Centre (RDC) of Statistics Canada. This study was exempt from ethics approval, since we used secondary data from a national survey conducted by Statistics Canada.

### 2.2. Analytical Sample 

This study included all individuals aged ≥1 year, excluding pregnant and lactating women and individuals with no dietary data, resulting in a final sample size of 20,176. Individuals more than 1 year of age were added to the analyses because, from this age, children are introduced to solid food and breastfeeding is often discontinued.

### 2.3. Plant-Based Food and Red/Processed Meat Dishes Categories

The red/processed meat dishes that we included in this study were any beef/pork/lamb/goat dishes. This category included a range of 265 unique food codes from the Canadian Nutrient File (CNF). Any red/processed meats that are not part of any dishes were not included. Plant-based food categories included nuts, seeds and nuts, seed mixes, trail mixes, plant-based beverages, nut butters, legume dishes with meat, legume dishes without meat, Mexican dishes, rice and rice mixed dishes, soups, canned/jarred vegetables and legumes, legumes, grains, tofu and meat substitutes and plant-based non-dairy desserts. Fruits and vegetables were excluded from this study given their low consumption and minor contribution to protein intakes. This category included a range of 659 unique food codes from the CNF.

### 2.4. Statistical Analyses

We identified the top 10 most frequently consumed plant-based foods and red/processed meat dishes among the Canadian population. If an individual reported any amount of a particular food item (i.e., if serving size > 0 of the specific food code), then it was considered as one eating occasion of that specific food code. If any individual has the same food more than once a day, that person contributed to more than one eating occasion. Among the top 10 most frequently consumed plant-based foods and red/processed meat dishes, we ranked the highest contributors of energy, protein, saturated fat, sodium, and sugars to the daily intake of Canadians. We reported the percentage of individuals consuming the top five plant-based foods and red/processed meat dishes. Further, the mean amount and the mean proportion of energy, protein, saturated fat, sodium, and sugars derived from the top five plant-based foods and red/processed meat dishes were reported per consumer. For the mean percent contribution, individual percentages were calculated first and then the average was taken of that individual’s percentages. All analyses were performed using SAS (version 9.3). We used appropriate bootstrapping weights to obtain population-level estimates using Statistics Canada guidelines [15]. The values were reported as mean ± SE or % ±SE for continuous or categorical variables, respectively.

## 3. Results

Table 1 represents the top 10 most frequently reported plant-based foods and red/processed meat dishes among Canadians aged ≥1 year along with the percentages of individuals reporting the food. The most frequently consumed red/processed meat dish was “Cooked regular, lean or extra lean ground beef or patty”. “Ham and cheese sandwich with lettuce and spread” and “ham sandwich with lettuce and spread” were the second and third most frequently consumed red/processed meat dishes. “Frankfurter (wiener/hot dog) on bun”, “homemade-style spaghetti sauce”, “barbecued pork”, and “shepherd’s pie” were also among the top 10 most frequently reported red/processed meat dishes. The most frequently consumed plant-based food in the Canadian population was “Cooked regular long-grain white rice”, followed by “canned red ripe tomatoes”. “Canned tomato puree, no salt added”, and “smooth type peanut butter, fat, sugar and salt added” were the third and fourth most frequently consumed plant-based foods. “Canned cream of asparagus soup, ready-to-serve beef or chicken broth soup”, “canned tomato sauce”, “sweetened enriched almond milk”, and “dried almonds” were also among the top 10 most frequently reported plant-based foods.

Ranking of the top 10 most frequently reported red/processed meat dishes based on their contribution to nutrients-to-limit, energy, and protein intake is presented in Table 2. Among the top 10 most frequently reported red/processed meat dishes, “ham and cheese sandwich with lettuce and spread” was the largest contributor to energy, protein, and saturated fat intake, and the second-largest contributor to sugars and sodium intake within the red/processed meat dishes category. Moreover, among red/processed meat dishes, “ham sandwich with lettuce and spread” was the top contributor to sodium intake, and “frankfurter (wiener/hot dog), with ketchup and/or mustard, on bun” was the top contributor to sugars intake in the Canadian diet within the red/processed meat dishes category.

Table 3 reports the ranking of the top 10 most frequently reported plant-based foods based on their contribution to energy, protein, saturated fat, sodium, and sugars intake. Among plant-based foods, “cooked regular long-grain white rice” was the largest contributor to energy and protein intake within the plant-based food category. In addition, “smooth type peanut butter with added fat, sugar and salt” was the top contributor to saturated fat intake, “ready-to-serve beef broth soup” was the top contributor to sodium intake, and “vanilla flavored sweetened enriched almond milk” was the top contributor to sugars intake among the top 10 most frequently reported plant-based foods within the plant-based food category.

Table 4 presents the proportion of individuals consuming the top five most frequently reported red/processed meat dishes as well as the mean and percentage contribution of each food item to energy, protein, saturated fat, sugars, and sodium intake of consumers on any given day. Since these results are based on 24 h recall, the proportion of individuals consuming the top five food sources are based on any given day. The proportion of Canadians consuming “cooked regular, lean or extra lean ground beef or patty” was 4.6%, followed by 2.9% for “ham sandwich with lettuce and spread”. Among red/processed meat dishes, “ham and cheese sandwich with lettuce and spread” made the largest contribution to energy (21.8%), protein (31.4%), saturated fat (31.0%), and sodium (41.8%) intake among its consumers. In addition, “frankfurter (wiener/hot dog), with ketchup and/or mustard, on bun” accounted for 10.9% of the sugars intake of the consumers of this food item.

As reported in Table 5, the proportion of Canadians consuming “cooked regular long-grain white rice” was 15.5%, followed by 13.4% for “canned red ripe tomato”. Among plant-based foods, “cooked regular long-grain white rice” made the largest contribution to energy intake (12.1%), followed by “smooth type peanut butter with added fat, sugar and salt” (8.4%). In addition, “smooth type peanut butter with added fat, sugar and salt” made the largest contribution to protein (8.1%), saturated fat (13.3%), sodium (4.8%), and sugars (3.8%) intake among its consumers.

## 4. Discussion

This is the first study to identify the top 10 most frequently consumed plant-based foods and red/processed meat dishes in the Canadian population, using a nationally representative sample. Overall, the most frequently consumed plant-based food in Canada was “cooked regular long-grain white rice”. The mean contribution of this food item to the energy intake of its consumers was just under 200 kcal/day. The most frequently consumed red/processed meat dish was “cooked regular, lean or extra lean ground beef or patty”. Among all red/processed meat dishes, “ham and cheese sandwich with lettuce and spread” made the largest contribution to energy, protein, saturated fat, and sodium intake.

Rice is the most widely consumed food staple for almost 50% of the world’s population [16]. On a global basis, rice provides 21% of energy and 15% of protein per capita [17], and long-grain white rice is known to be the most commonly eaten form of rice [18]. Similarly, our results revealed that the most frequently consumed plant-based food in Canada in 2015 was “cooked regular long-grain white rice”. More than 15% of Canadians reported consuming this food item, and it contributed to 12.1% of energy intake and 6.1% of protein intake among consumers. Using data from the CCHS 2015, Kirkpatrick et al. reported that rice and rice mixed dishes were among the top 10 contributors to energy intake in the general Canadian population (≥1 year), and among the top five contributors to energy intake in the low-income group [6]. In a study investigating the dietary sources of energy and nutrient intakes among five ethnic groups (i.e., Caucasian; Latino; Native Hawaiian; Japanese-American; African American) in the U.S., it was found that rice made a significant contribution to dietary energy intake, ranging from 5.3% (Caucasian women) to 22.9% (Japanese-American men). In addition, rice was found to be the top dietary source of protein for Japanese-American men and women, respectively, contributing 12.7% and 10.4% to protein intake [19]. Sharma et al. also reported that white rice was the most commonly consumed grain food among Japanese-American, Native Hawaiian, and Caucasian men (12.0–44.1%), and the most commonly consumed refined grain among all ethnic groups (10.3–54.1%), except for Latinos [20]. In another study aiming to describe the most commonly consumed foods in the Brazilian diet, Souza et al. found that rice (84%) was the most frequently recorded food by Brazilian individuals (≥10 years), followed by coffee (79%) and beans (72.8%) [7]. Using data from the 2009–2013 Korea National Health and Nutrition Examination Surveys (KNHANES), it was revealed that white rice was the major source of energy (31%) among Korean preschoolers aged 1–5 years, followed by milk (10.2%) and bread (3.5%) [21]. Although white rice (milled and/or polished rice) is the most commonly consumed form of rice, it is a poor source of vitamins and minerals due to removing the bran and germ layers of the seed during the milling process [22]. In their review of the literature, Saleh et al. reported that brown rice is nutritionally superior to white rice because of higher levels of nutrients such as protein, vitamins, and minerals [22]. In accordance with this, the new Canada’s Food Guide has placed a major emphasis on whole grain foods and recommended regular consumption of whole grains and decreasing consumption of refined grains [5].

Our results showed that the most frequently consumed red/processed meat dish in Canada in 2015 was “cooked regular, lean or extra lean ground beef or patty”. This food item was consumed by 4.6% of the Canadian population and accounted for 4% of energy intake, 9.5% of saturated fat intake, and 10.5% of protein intake among its consumers. Ground beef is the most commonly consumed form of beef in the U.S., accounting for more than 40% of all beef consumed [23]. Results from a multiethnic cohort study of 215,000 individuals aged 45–75 years showed that lean beef (steak/roast) was the most commonly consumed red meat for all ethnic groups in the U.S. (9.3–14.3%), except for Japanese-Americans and Native Hawaiians [24]. Using data from the 2003–2006 National Health and Nutrition Examination Survey (NHANES), O’Neil et al. reported that beef was the top source of protein (14.0%), the third-highest ranked source of saturated fat (7.9%), and the fourth-highest ranked source of energy (5.0%) among American adults aged ≥19 years [25]. Using the same survey data, Huth et al. found that the top three food sources of saturated fat in the diet of Americans aged ≥2 years were cheese (16.5%), beef (8.5%), and milk (8.3%) [26]. In addition, results from the 2007–2010 NHANES showed that ground beet contributed to 5.6% of animal protein intake and 2.6% of total protein intake among U.S. adults aged 19 years and older [27]. Souza et al. reported that beef was the fifth most commonly reported food item, and the most frequently reported animal-based food on the first day’s food record of Brazilian individuals (≥10 years) [7]. A more recent study also showed that beef was the most commonly consumed meat among Brazilian individuals aged 10 years and older (49%), and the mean beef intake for the entire country was 63 g/day [28]. In a survey conducted in Spanish adults aged 25 to 75 years, beef was found to be the most frequently consumed red meat (63.6%), with pork in second place (52.6%) [29]. Using data from the 2011–2012 National Nutrition and Physical Activity Survey (NNPAS), Sui et al. reported that red meat was consumed by 48.6% of Australian men and women, with beef as the most frequently reported type (41.8% and 34.7%, respectively) [30]. Some observational studies have found an association between consumption of beef and increased risk of a number of cancers [31,32,33,34]. Therefore, reducing beef intake and replacing it with plant-based proteins could be an effective strategy in the prevention of such conditions. In line with this, the new Canada’s Food Guide has recommended regular consumption of protein foods, with particular emphasis on plant-based proteins [5]. Further, according to the new food guide, ruminant animal-based foods such as beef and lamb are natural sources of trans fat, a type of unsaturated fat known to have adverse health effects [5].

Processed meats, such as ham, bacon, frankfurters, and sausages that have been modified through salting, curing, fermentation, or smoking, account for a large proportion of the world’s meat consumption [35]. Our results revealed that “ham and cheese sandwich with lettuce and spread” was the second most frequently consumed red/processed meat dish in Canada in 2015. This food item accounted for 41.8% of sodium intake, 31.4% of protein intake, 31.0% of saturated fat intake, and 21.8% of energy intake among its consumers. Among red/processed meat dishes, “ham sandwich with lettuce and spread” and “frankfurter (wiener/hot dog) with ketchup and/or mustard on bun” were in the next rankings. Using data from the 2009–2010 NHANES, Sebastian et al. reported that nearly half of American adults (49%) reported eating a sandwich on any given day, and the mean contribution of sandwiches to energy intakes of all adults was 200 kcal for women and 350 kcal for men [36]. In addition, the mean contribution of sandwiches to sodium intake of American men and women was 902 mg and 489 mg, respectively [36]. In the U.S., processed meat intake constitutes 22% of the total meat consumed from either poultry or red meat categories [37]. Using data of Australians aged ≥2 years from the 2011–2012 NNPAS, Sui et al. found that 37.8% of participants reported consuming processed meat, and ham was the most frequently reported type of processed meat (females 16.8%, males 19.4%), followed by bacon (females 12.4%, males 15.3%), and sausage (females 5.8%, males 8.5%) [30]. Kirkpatrick et al. reported that processed meats are among the top 10 Contributors to sodium and saturated fat intake of Canadians aged 1 year and older [6]. In a study aiming to identify the major food sources of dietary sodium using 3-day food records, it was found that processed meat was the largest contributor to daily sodium intake among Mexican adults, representing 8% of total sodium intake per capita [38]. The results also showed that total sodium contributed by processed meat in the entire sample population was 223 mg/day [38]. Using data obtained from 21,108 British households, Ni Mhurchu et al. observed that processed meat (18%) was the second largest contributor to sodium purchases after table salt (23%) [39]. In a nationwide survey conducted in Brazil, it was found that processed meat was among the top five contributors to saturated fat intake among Brazilian individuals aged ≥10 years [40]. Moreover, increased morbidity and mortality related to high consumption of processed meat has been linked to their high content of sodium and saturated fat [41,42,43,44]. We have recently shown that reducing red and processed meat by half and increasing plant-based meat alternatives by 100% may assist in reducing the intake of sodium and saturated fat, and increase the overall nutritional value of the diet [45].

### Strengths and Limitations

This study used nutrition data from CCHS–Nutrition 2015, a nationally representative survey of the Canadian population aged one year and older. A major strength of this study was the opportunity to identify the top 10 most frequently consumed plant-based foods and red/processed meat dishes, shortly after the introduction of the new Canada’s Food Guide, which places a major emphasis on the consumption of plant-based foods. We also acknowledge some limitations. First, since we combined age and sex groups in this study, some results may vary for different age/sex groups. Second, detailed dietary data were obtained using a 24 h dietary recall and, therefore, may not reflect the usual intake of the participants. In addition, the 24 h dietary recall is a self-report method subject to recall bias and misreporting (i.e., overestimating or underestimating dietary intake). Third, CCHS–Nutrition 2015 does not include added sugar intake information of Canadians. That is why we were only able to report the total sugar intake for the analyses. Fourth, the descriptive, cross-sectional design of the study does not allow us to determine the causal relationship between patterns of food consumption and health outcomes. Fifth, we included the consumption data of only two main food categories, namely plant-based foods and red/processed meat dishes. It is notable that the plant-based foods make up a broader category compared to the red/processed meat dishes category, and include vegetables and fruit, grain products, and protein-based foods.

## 5. Conclusions

The results of this study revealed that the most frequently consumed plant-based food among the Canadian population aged ≥1 year was “cooked regular long-grain white rice”, followed by “canned red ripe tomatoes”. Cooked regular long-grain white rice made a significant contribution to energy and protein intake among the consumers of this food item. Among the top five most frequently consumed plant-based foods, “smooth type peanut butter, fat, sugar, and salt added” made the most significant contribution to saturated fat, protein, sugars, and sodium intake. The most frequently consumed red/processed meat dish in Canada was “Cooked regular, lean or extra lean ground beef or patty”. Further, the top five red/processed meat dishes, especially “ham and cheese sandwich with lettuce and spread”, made a significant contribution to the intake of saturated fat, sugars, and sodium in the diet of Canadians. According to the new Canada’s Food Guide, patterns of eating that place a major emphasis on plant-based foods typically result in higher intakes of dietary fiber, nuts, vegetables and fruit, and soy protein, and also encourage lower intakes of processed meat and saturated fat [5]. Thus, minimizing the consumption of red/processed meat dishes and shifting intakes towards more plant-based foods may improve the health of Canadians and confer environmental benefits [5,11]. Putting this into practice, in the most recent Canada’s Food Guide, Health Canada has merged the two food groups (Meat & Alternatives and Milk & Alternatives) of the 2007 Food Guide into a single group called “Protein Foods” [5]. Among protein foods, health Canada also recommended consuming plant-based more often. The new guidelines along with relevant health promotion initiatives may have an impact on shifting toward consumption of more plant-based foods among Canadians. Further research is required to determine and compare the most frequently consumed red/processed meat dishes and plant-based foods among different age/sex groups of Canadians and how they can contribute to the intake of energy, protein, and nutrients-to-limit.

## Figures and Tables

**Table 1 nutrients-14-01257-t001:** Top 10 most frequently reported plant-based foods and red/processed meat dishes among individuals ≥ 1 year in Canada (*n* = 20,176), 2015 Canadian Community Health Survey.

Rank	Red/Processed Meat Dishes	% of Individuals Reporting the Red/Processed Meat Dishes	Plant-Based Foods	% of Individuals Reporting the Plant-Based food
1	Ground beef or patty, cooked, NS as to regular, lean, or extra lean	4.6	Grains, rice, white, long-grain, regular, cooked	15.5
2	Sandwich, ham and cheese, with lettuce and spread	2.6	Tomato, red, ripe, canned, whole	13.4
3	Sandwich, ham, with lettuce and spread	2.9	Tomato products, canned, puree, no salt added	10.1
4	Frankfurter, wiener, or hot dog, with ketchup and/or mustard, on bun	2	Peanut butter, smooth type, fat, sugar and salt added	10.2
5	Frankfurter, wiener, or hot dog, plain, on bun	1.1	Soup, canned, cream of asparagus, condensed, milk added	7.5
6	Spaghetti sauce with beef or meat other than lamb or mutton, homemade-style	1.7	Soup, broth, beef, ready-to-serve	6.6
7	Spaghetti sauce with meat and vegetables, homemade-style	1.2	Tomato products, canned, sauce	5.9
8	Spaghetti sauce with a combination of meats, homemade-style	1.1	Soup, broth, chicken, ready-to-serve	4.8
9	Pork, spareribs, barbecued, with sauce, NS as to fat eaten	0.8	Plant-based beverage, almond, enriched, sweetened, vanilla flavored	2.8
10	Shepherd’s pie, with corn	0.8	Nuts, almonds, dried, unblanched, unroasted	3.3

NS: Not specified.

**Table 2 nutrients-14-01257-t002:** Ranking of the top 10 most frequently reported red/processed meat dishes based on their contribution to nutrients-to-limit, energy, and protein intake (*n* = 20,176), 2015 Canadian Community Health Survey.

	Red/Processed Meat Dishes	Dietary Components and Rank				
		Energy	Protein	SFA	Sugars	Sodium
1	Ground beef or patty, cooked, NS as to regular, lean, or extra lean	5	3	3	10	9
2	Sandwich, ham and cheese, with lettuce and spread	1	1	1	2	2
3	Sandwich, ham, with lettuce and spread	2	2	4	3	1
4	Frankfurter, wiener, or hot dog, with ketchup and/or mustard, on bun	3	4	2	1	3
5	Frankfurter, wiener, or hot dog, plain, on bun	6	9	6	9	7
6	Spaghetti sauce with beef or meat other than lamb or mutton, homemade-style	8	6	8	4	4
7	Spaghetti sauce with meat and vegetables, homemade-style	10	10	10	7	5
8	Spaghetti sauce with combination of meats, homemade-style	9	8	9	5	6
9	Pork, spareribs, barbecued, with sauce, NS as to fat eaten	7	7	5	6	10
10	Shepherd’s pie, with corn	4	5	7	8	8

NS: Not specified; SFA: Saturated fatty acids.

**Table 3 nutrients-14-01257-t003:** Ranking of the top 10 most frequently reported plant-based foods based on their contribution to nutrients-to-limit, energy, and protein intake (*n* = 20,176), 2015 Canadian Community Health Survey.

	Plant-Based Foods	Dietary Components and Rank				
		Energy	Protein	SFA	Sugars	Sodium
1	Grains, rice, white, long-grain, regular, cooked	1	1	4	8	8
2	Tomato, red, ripe, canned, whole	7	7	5	4	5
3	Tomato products, canned, puree, no salt added	6	6	7	3	7
4	Peanut butter, smooth type, fat, sugar and salt added	2	2	1	2	4
5	Soup, canned, cream of asparagus, condensed, milk added	3	5	2	9	9
6	Soup, broth, beef, ready-to-serve	9	4	8	6	1
7	Tomato products, canned, sauce	8	8	6	5	3
8	Soup, broth, chicken, ready-to-serve	10	9	9	10	2
9	Plant-based beverage, almond, enriched, sweetened, vanilla flavored	5	10	10	1	6
10	Nuts, almonds, dried, unblanched, unroasted	4	3	3	7	10

SFA: Saturated fatty acids.

**Table 4 nutrients-14-01257-t004:** The mean and percentage contribution of the top five reported red/processed meat dishes to nutrients-to-limit, energy, and protein intake of Canadians aged ≥1 year in Canada, 2015 Canadian Community Health Survey.

	% Population (% ±SE)	Dietary Components ^1^				
Top 5 Red/Processed Meat Dishes		Energy	Protein	SFA	Sugars	Sodium
# 1 Ground beef or patty, cooked, NS as to regular, lean, or extra lean	4.6 ± 0.4					
Daily intake (mean ± SE)		74.7 ± 3.57	7.9 ± 0.4	1.8 ± 0.1	0	116.2 ± 5.6
% Contribution per consumer/day (% ±SE)		4.0 ± 0.2	10.5 ± 0.5	9.5 ± 0.6	0	4.3 ± 0.2
# 2 Sandwich, ham and cheese, with lettuce and spread	2.6 ± 0.2					
Daily intake (mean ± SE)		421.5 ± 14.9	24.8 ± 0.9	7.8 ± 0.4	5.8 ± 0.2	1408.5 ± 54.6
% Contribution per consumer/day (% ±SE)		21.8 ± 0.7	31.4 ± 1.1	31.0 ± 1.0	8.2 ± 0.5	41.8 ± 1.3
# 3 Sandwich, ham, with lettuce and spread	2.9 ± 0.3					
Daily intake (mean ± SE)		332.9 ± 18.6	19.2 ± 1.0	3.3 ± 0.4	5.6 ± 0.3	1226.7 ± 59.9
% Contribution per consumer/day (% ±SE)		17.7 ± 0.8	25.2 ± 1.1	14.6 ± 1.0	7.1 ± 0.5	38.0 ± 1.3
# 4 Frankfurter, wiener, or hot dog, with ketchup and/or mustard, on bun	2.0 ± 0.2					
Daily intake (mean ± SE)		351.7 ± 19.2	14.0 ± 0.8	5.8 ± 0.3	8.3 ± 0.5	1154.5 ± 51.2
% Contribution per consumer/day (% ±SE)		18.1 ± 0.8	20.6 ± 1.1	24.0 ± 1.1	10.9 ± 0.9	36.9 ± 1.3
# 5 Frankfurter, wiener, or hot dog, plain, on bun	1.1 ± 0.2					
Daily intake (mean ± SE)		257.8 ± 15.9	10.9 ± 0.6	5.3 ± 0.4	3.9 ± 0.5	844.4 ± 50.8
% Contribution per consumer/day (% ±SE)		13.7 ± 0.8	16.4 ± 1.2	22.4 ± 0.2	6.5 ± 1.5	27.6 ± 1.3

NS: Not specified; SFA: Saturated fatty acids. ^1^ Units: Energy in kcal, Protein in grams, SFA in grams, Sugars in grams, and Sodium in mg.

**Table 5 nutrients-14-01257-t005:** The mean and percentage contribution of the top five reported plant-based foods to nutrients-to-limit, energy, and protein intake of Canadians aged ≥1 year in Canada, 2015 Canadian Community Health Survey.

	% Population (% ±SE)	Dietary Components ^1^				
Top 5 Plant-Based Foods		Energy	Protein	SFA	Sugars	Sodium
# 1 Grains, rice, white, long-grain, regular, cooked	15.5 ± 0.6					
Daily intake (mean ± SE)		199.4 ± 5.8	4.1 ± 0.1	0.1 ± 0.0	0.1 ± 0.0	1.5 ± 0.04
% Contribution per consumer/day (% ±SE)		12.1 ± 0.3	6.1 ± 0.2	0.9 ± 0.04	0.2 ± 0.0	0.09 ± 0.0
# 2 Tomato, red, ripe, canned, whole	13.4 ± 0.5					
Daily intake (mean ± SE)		7.6 ± 0.3	0.4 ± 0.01	0.02 ± 0.0	1.2 ± 0.1	54.3 ± 2.2
% Contribution per consumer/day (%)		0.3 ± 0.01	0.5 ± 0.02	0.07 ± 0.0	1.7 ± 0.1	1.9 ± 0.07
# 3 Tomato products, canned, puree, no salt added	10.1 ± 0.4					
Daily intake (mean ± SE)		14.6 ± 0.7	0.6 ± 0.03	0.01 ± 0.0	1.9 ± 0.09	10.7 ± 0.5
% Contribution per consumer/day (% ±SE)		0.7 ± 0.03	0.9 ± 0.03	0.05 ± 0.0	0.03 ± 0.01	0.3 ± 0.01
# 4 Peanut butter, smooth type, fat, sugar and salt added	10.2 ± 0.5					
Daily intake (mean ± SE)		168.1 ± 8.5	6.2 ± 0.3	2.8 ± 0.2	2.9 ± 0.2	119.8 ± 6.1
% Contribution per consumer/day (% ±SE)		8.4 ± 0.03	8.1 ± 0.3	13.3 ± 0.05	3.8 ± 0.2	4.8 ±0.2
# 5 Soup, canned, cream of asparagus, condensed, milk added	7.5 ± 0.4					
Daily intake (mean ± SE)		45.4 ± 3.1	1.1 ± 0.1	0.4 ± 0.03	0.2 ± 0.01	0.2 ± 0.01
% Contribution per consumer/day (% ±SE)		2.5 ± 0.2	1.6 ± 0.1	2.3 ± 0.2	0.3 ± 0.03	0.01 ± 0.0

SFA: Saturated fatty acids. ^1^ Units: Energy in kcal, Protein in grams, SFA in grams, Sugars in grams, and Sodium in mg.

## Data Availability

The data will be available at the Research Data Centre (RDC) of Statistics Canada.

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
