# Peer review of "Most Frequently Consumed Red/Processed Meat Dishes and Plant-Based Foods and Their Contribution to the Intake of Energy, Protein, and Nutrients-to-Limit among Canadians"

_nutrients, 2022, doi:10.3390/nu14061257_

Round 1
Reviewer 1 Report
Dear Authors,
congrats to an overall already good and intersting paper - i enjoyed the read!
The major point to clarify and fix is age and sex groups to be pooled and if really meaningful - I guess not - and if the starting age of 1 yr makes sense! With this, also the statistical approach/modelling has to be reflected once again -> See my comments on this on the basis of different energy and nutritional requirements through the phases of life cycle as well as between sexes. You list this as limitaton, yes, but is the major point of criticism of this paper as I think can be much better paper with analysis by age groups and sexes IN ADDITION
Formatting:
Tables are not well designed and shall be presented better (eg. long variable/paramter text in colum 1 or 2 must be avoided); especially Tables 4 + 5 are confusing and crushing
spaces between last word and reference brackets all through the text!
All good success!
Reviewer 2 Report
Dear Authors,
Thank you for the opportunity to review this research dealing with Canadians foods habits.
I have some minor concerns which I have to address you:
The title is too broad: please change it to better highlights your interesting research in less words.
The literature you cite is not updated (only 1 manuscript refers to 2020, no manuscript published in 2021). We are in 2022! Please provide.
The conclusion have to better highlight the results and please let us know if further researches are planned.
Best regards
